# The Structural and Electronic Properties of the Ag$_5$ Atomic Quantum Cluster Interacting with CO$_2$, CH$_4$, and H$_2$O Molecules

Moteb Alotaibi [1],*, Turki Alotaibi [2], Majed Alshammari [2] and Ali K. Ismael [3],*

1 Department of Physics, College of Science and Humanities in Al-Kharj, Prince Sattam bin Abdulaziz University, Al-Kharj 11942, Saudi Arabia
2 Physics Department, College of Science, Jouf University, Sakakah 11942, Saudi Arabia
3 Department of Physics, Lancaster University, Lancaster LA1 4YB, UK
* Correspondence: mot.alotaibi@psau.edu.sa (M.A.); k.ismael@lancaster.ac.uk (A.K.I.)

**Abstract:** Recent advancements in experimental approaches have made it possible to synthesize silver (Ag$_5$) atomic quantum clusters (AQCs), which have shown a great potential in photocatalysis. This study employs the generalized gradient approximation (GGA) density functional theory (DFT) to explore the adsorption of CO$_2$, CH$_4$, and H$_2$O molecules on the Ag$_5$ AQC. Our investigations focus on the structural and electronic properties of the molecules in Ag$_5$ AQC systems. This involves adsorption energy simulations, charge transfer, charge density difference, and the density of states for the modelled systems. Our simulations suggest that CH$_4$ and H$_2$O molecules exhibit higher adsorption energies on the Ag$_5$ AQC compared to CO$_2$ molecules. Remarkably, the presence of CH$_4$ molecule leads to a significant deformation in the Ag$_5$ AQC structure. The structure reforms from a bipyramidal to trapezoidal shape. This study also reveals that the Ag$_5$ AQC donates electrons to CO$_2$ and CH$_4$ molecules, resulting in an oxidation state. In contrast, gaining charges from H$_2$O molecules results in a reduced state. We believe the proposed predictions provide valuable insights for future experimental investigations of the interaction behaviour between carbon dioxide, methane, water molecules, and Ag$_5$ sub-nanometre clusters.

**Keywords:** Ag$_5$ atomic quantum cluster; DFT; renewable energy

## 1. Introduction

Metal nanoparticles with sizes ranging from 2 to 100 nm have been extensively studied and applied in various fields such as catalysis [1], bioassays [2], and medicine [3]. When the size of metal particles is reduced to approximately 0.5 nm for silver (Ag) or gold (Au) [4], new chemical, optical, and electrical attributes appear, which differ significantly from bulk and nanomaterials. These metal particles are known as atomic quantum clusters (AQCs) and possess discrete electronic energy levels due to their very small sizes. Surprisingly, these metal AQCs are highly stable due to strong quantum confinement, which includes the highest occupied molecular orbital (HOMO) and lowest unoccupied molecular orbital (LUMO) gap at the Fermi energy level [5]. As the size of the metal clusters reduces, the HOMO–LUMO gap increases, leading to a reduction in its reactivity. Currently, small metal clusters with a size of less than 1 nm are being synthesized and widely studied under realistic reaction conditions [6–8]. Additionally, experimental investigations have revealed that these sub-nanometre clusters have unique catalytic activities due to their high surface-to-volume ratio and high concentration of impurity atoms compared with the bulk materials [9–12]. Silver AQCs play a pivotal role in various technological processes, including photocatalysis, where they are utilized as cocatalysts.

In recent years, the advancement of the density functional theory (DFT) and improvements in computer hardware have made quantum mechanical calculations and molecular modelling techniques effective tools for investigating the adsorption mechanism of silver

clusters on metal oxide surfaces [13,14]. Computational techniques have been widely employed to gain mechanistic insights into metallic nanoparticles, which are highly attractive materials. In particular, Hagen et al. [15] provided theoretical evidence that odd-sized Ag cluster anions, specifically $Ag_3$ and $Ag_5$, can activate the molecular bond of oxygen through cooperative effects in the adsorption of two oxygen molecules. Furthermore, it has been found that $Ag_5$ AQCs exhibit exceptional catalytic activities upon the adsorption on the rutile $TiO_2$ (110) surface. Alotaibi et al. [16], for instance, conducted a systematic study on the impact of the $Ag_5$ AQC on the photocatalytic activities of rutile and anatase $TiO_2$ using DFT calculations. The deposition of $Ag_5$ on titania surfaces was predicted to induce electronic gap states due to charge transfers from the cluster, resulting in the enhanced photocatalytic activities of the material. Gogoi et al. [17] observed a significant reduction in the band gap of a $TiO_2$ surface upon depositing 1.5 Ag, leading to an improvement in the photocatalytic activity for hydrogen production. Additionally, Preda et al. [18] employed DFT to investigate the Ag cluster on pure and defective $CeO_2$ (111) surfaces. It was found that the Ag cluster undergoes oxidation, transferring its valence electrons to the $CeO_2$ surface, thereby enhancing its photocatalytic activity.

Several theoretical and experimental studies have been conducted on the mechanism of silver clusters. However, the important mechanisms necessary for fabricating higher levels of photocatalytic materials have not been thoroughly investigated either theoretically or experimentally. Notably, the literature lacks calculations of $Ag_5$ AQCs with $CO_2$, $CH_4$, and $H_2O$ molecules, and investigations of their geometries and electronic properties are of great significance, given that they can now be synthesized and produced without ligands [19,20]. The combustion of fossil fuels, namely natural gas, oil, and coal is a significant contributor to the emission of greenhouse gases. The release of large quantities of $CO_2$ and $CH_4$, among other gases, has a detrimental impact on the environment. Methane leakage, for instance, represents a considerable waste of resources and poses significant pollution issues. Given the pressing need for clean energy, it is of utmost importance to develop efficient hydrogen evolution reaction (HER) catalysts [21] that can split water into oxygen and hydrogen. Therefore, it is crucial to identify functionally tailored materials that can enable cross-selectivity techniques for the development of nonconventional, cleaner energy sources.

This paper aims to investigate the interactions between $Ag_5$ AQCs with $CO_2$, $CH_4$, and $H_2O$ molecules, by considering their corresponding geometries and electronic structures, using DFT. The materials presented in this paper have the potential for various technological applications, including hydrogen production, methane gas sensors, and nanoelectronics. To explore the interactions of $Ag_5$ AQCs with these molecules, we have computed different electronic characteristics, including the adsorption energy, charge transfer using Bader charge analysis, charge density difference, and density of states (DOSs). This study could serve as a valuable theoretical guide for future experimental work towards photocatalytic processes used in the field of renewable energy. The structure of this paper is as follows. In Section 2, we elaborate on the computational methodologies developed and utilized in this study. Subsequently, the findings from our computations are expounded upon and deliberated in Section 3, which is divided into several subsections. Finally, Section 4 summarizes our primary results, offering a pragmatic outlook on their implications.

## 2. Computational Details

In this study, quantum chemical simulations were implemented to investigate the structural and electronic properties of the $Ag_5$ AQC and its interaction with the studied molecules. The Kohn–Sham DFT technique [22,23] was used, employing the Quantum Espresso simulation package v6.7 [24]. The exchange correlation functionals were treated within the generalised gradient approximation (GGA) suggested by Perdew-Burke-Ernzerhof (PBE) [25]. A plane-wave basis set with an energy cut-off value of 25 Ry for the wave function and 225 Ry for the charge density were fixed for all calculations present in this study. We also tested the optimisation results by increasing the energy cut-off value to 50 Ry, and found the same optimised configurations revealed by the initial energy

cut-off value. See Figure S9 for comparison purposes. The projector augmented wave (PAW) method [26,27] pseudopotentials were employed to describe the interaction between valence electrons and the ion core. The PAW technique allowed for the precise treatment of the valence band states.

All modelled systems were placed in sufficiently large simple cubic cells with a side length of 15 Å to prevent any interactions between molecules in different replicas. To account for the large size of the unit cells, all simulations were restricted to the Γ-point within a Brillouin zone. During the geometric optimization, a convergence threshold value of $10^{-6}$ Ry was utilized for the self-consistency electronic minimization, and all atoms were allowed to relax with a threshold force value of $10^{-3}$ Ry/Å. To analyse electron transport, Bader charge analysis was employed [28]. The VESTA software [29] was used to analyse maps of charge density difference, and to visualise all structures. It is important to note that all simulations utilized spin-polarized calculations to obtain accurate electronic structures. The adsorption energy ($E_{ads}$) of the $CO_2$, $CH_4$, and $H_2O$ molecules on the $Ag_5$ AQC was calculated using the formula provided below.

$$E_{ads} = E_{tot} - E_{Ag_5} - E_{molecule} \tag{1}$$

In the equation, $E_{tot}$ represents the total energy of the entire system, $E_{Ag_5}$ denotes the total energy of the bare $Ag_5$ cluster, and $E_{molecule}$ signifies the total energy of the isolated molecule. The subsequent section undertakes an examination of the optimised structures and adsorption energies. Furthermore, an exploration into the influence of $CO_2$, $H_2O$, and $CH_4$ on the electronic structures of the $Ag_5$ AQC is conducted through the presentation of the density of states, Bader charge analysis, and charge density difference.

## 3. Results and Discussion

The structure and electronic properties of the $Ag_5$ atomic quantum cluster interacting with $CO_2$, $CH_4$, and $H_2O$ molecules were explored as follows.

### 3.1. The CO₂@Ag₅ Structure

We started the calculations by examining the geometrical and electronic properties of the isolated systems. For example, the isolated $Ag_5$ nanocluster, represented by a doublet with S = 1/2 in the density of state plot (see Figure S4), has an unpaired electron that is predominantly located on the two axial Ag atoms [16]. This unpaired electron is represented by the singly occupied molecular orbital (SOMO), which is suited right below the Fermi energy level. This distribution of charge is similar to that observed in $Cu_5$ [30]. Furthermore, the existence of the unpaired electron in the $Ag_5$ nanocluster can influence its electronic properties. Unpaired electrons are often involved in the electronic and optical attributes of materials. For instance, Yang et al. [31] conducted a theoretical and experimental study on Cu or Zn on metal-organic frameworks (MOFs) containing electron-withdrawing ligands. Their results revealed that electrons induced wide-range light absorption within the MOFs, resulting in an excellent photo-induced intramolecular charge transfer (ICT) effect. The study verified the role of unpaired electrons in enhancing ICT within the MOFs through molecular structure, density of states, and electronic structure calculations, as well as the electron spin resonance (EPR) approach.

We investigated the interaction mechanism of the $CO_2$ molecule with the silver AQC by exploring its geometric optimization and electronic properties, including adsorption energy, charge transfer, charge density difference, and the density of states, to gain a deep understanding of the process. Three different adsorption sites of $CO_2$ on the $Ag_5$ cluster have been explored to find the most stable structure (see Figures 1 and S6). The initial and optimized structures of the most stable configuration of $CO_2$@$Ag_5$ are presented in Figure 1, which clearly shows that the $CO_2$ molecule was initially connected with two silver atoms (see Figure 1a). However, after the geometric optimization, only one oxygen atom was bonded to one silver atom, with a bond length of 2.56 Å (see Figure 1b and Table 1). The calculated distance between the carbon atom and the nearest silver atom was 3.25 Å,

which is consistent with a previous DFT study [32]. Meanwhile, a low stability was found for the structures possessing no bond between the oxygen atom and the silver atom. See Figure S6 and Table S8 for further details. The calculations of isolated systems, along with coordinates for all studied structures, can be found in the Supplementary Information.

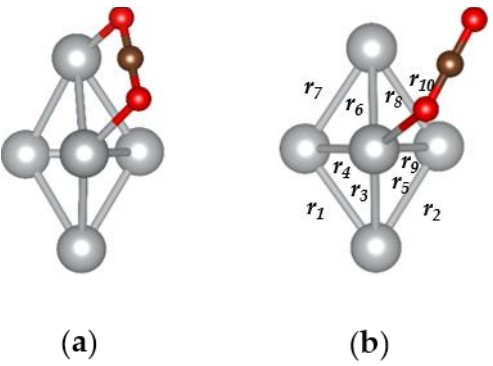

**Figure 1.** (**a**) Initial and (**b**) optimised structures of $CO_2$@$Ag_5$. Grey, red, and brown circles represent Ag, O, and C atoms, respectively.

**Table 1.** Bond lengths of $CO_2$@$Ag_5$ structure obtained using the PBE-DFT level.

| Symbol | Bond Length (Å) |
|---|---|
| $r_1$ | 2.82 |
| $r_2$ | 2.80 |
| $r_3$ | 2.76 |
| $r_4$ | 2.67 |
| $r_5$ | 2.72 |
| $r_6$ | 2.81 |
| $r_7$ | 2.81 |
| $r_8$ | 2.80 |
| $r_9$ (Ag-O) | 2.56 |
| $r_{10}$ (Ag-C) | 3.25 |

It is worth noting that upon the adsorption of the $CO_2$ molecule, the Ag–Ag bonds of the $Ag_5$ AQC were slightly increased, resulting in changes to the geometric structures of the silver cluster. Further details can be found in Table 1. These changes in the silver geometric structures when interacting with $CO_2$ molecules were also reported in a recent theoretical investigation [33]. Consequently, this increase in Ag–Ag bonds could enhance the surface activity of the silver cluster [34]. The adsorption energy, Eads, was calculated using the PBE-DFT level of the theory. The results indicated that the $CO_2$ molecule was physiosorbed to the $Ag_5$ cluster with an adsorption energy of −0.27 eV. This value is comparable to the results previously obtained by Zhang et al. [35]. Furthermore, it is important to note that the chemisorption feature of $CO_2$ on silver surfaces was not predicted in previous DFT simulations [36,37]. This is due to the unusual and unstable structure deformation of the $CO_2$ molecule. The calculated bond lengths of the bare $Ag_5$ cluster are illustrated in Figure S1d, which can be compared with the results presented in Table 1.

To investigate the charge transfer between the $Ag_5$ AQC and the $CO_2$ molecule, we conducted Bader charge analysis calculations [38]. The results of the Bader charge distribution revealed a modest charge donation from the silver cluster to the $CO_2$ molecule, amounting to approximately 0.004 e⁻. This charge transfer indicates that the cluster was oxidized. Additional information regarding the Bader charge analysis can be found in Table S5 of the Supplementary Information. This result aligns with a previous study conducted by Zhang et al. [34]. In their research, they employed a natural bond orbital (NBO) charge analysis to identify electronic transfer. Their results predicted that the $Ag_2$ cluster transfers approximately 0.009 e⁻ to the adsorbed $CO_2$ molecule. Furthermore, to

gain insights into the chemical bonds [39,40] and electronic structures of the $CO_2$@$Ag_5$ system, we examined its charge density difference and density of states, as depicted in Figure 2. The charge density difference plot reveals that the oxygen atoms of the $CO_2$ molecule exhibit negative surface potentials (represented by the blue clouds), while the silver atoms possess positive surface potentials (represented by the yellow clouds). This suggests that the interaction between the silver AQC and the $CO_2$ molecule is primarily driven by electrostatic forces.

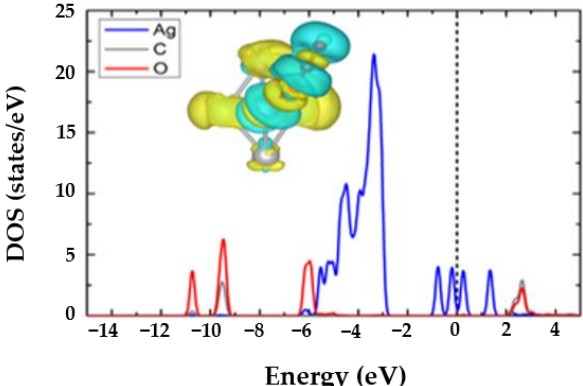

**Figure 2.** Charge density difference and density of states of $CO_2$@$Ag_5$. The blue, red, and grey curves of DOS represent the contributions of silver, oxygen, and carbon orbitals, while yellow and cyan clouds represent the positive and negative phases of charge density difference. (Note: The same reference colours are used for all the charge density difference plots in the subsequent figures.). The vertical dotted line refers to the Fermi energy level.

Furthermore, we conducted an analysis of the interaction between the $Ag_5$ AQC and the adsorbed molecule, focusing on the orbital overlaps. Our investigation revealed that the density of states exhibited the formation of four peaks (see Figure 2), which were generated successively as a result of the combination of the orbital of the C atom with that of the O atom. The DOS figure indicated a strong overlapped state of the orbital of the C atom and the orbital of the O atom, whose peak was located at approximately $-9.5$ eV. This observation suggests a significant electron transfer from the C atom to the O atoms, estimated to be around 2.16 $e^-$. It is worth noting that the energy level states of the $CO_2$ molecule experienced a significant shift of approximately $-3$ eV due to the interaction of the $Ag_5$ AQC (see Figures S2 and S4 of the DOS of the isolated $CO_2$ molecule and bare $Ag_5$ cluster for comparison purposes, respectively). Additionally, we observed a weak orbital overlap in low-energy states, which were located at less than $-6$ eV, for the binding interaction of the Ag atom with the O atom.

### 3.2. The $CH_4$@$Ag_5$ Structure

Three different adsorption sites of $CO_2$ on the $Ag_5$ cluster have been explored to find the most stable structure (see Figures 3 and S7). The structures presented in Figure S7 showing meta-stable states and the clusters retain their initial configurations, while Figure 3 illustrates the initial (bipyramidal) and optimized (trapezoidal) $Ag_5$ cluster configurations of the most stable $CH_4$@$Ag_5$ structure. Surprisingly, upon geometric optimization, a significant deformation of the silver cluster occurred when interacting with the methane molecule. This deformation resulted in a transition from a three-dimensional (bipyramidal) shape (see Figure 3a) to a two-dimensional (trapezoidal) shape, as depicted in Figure 3b. A previous DFT investigation [16] has reported that the trapezoidal-shaped $Ag_5$ cluster is energetically favourable compared to the bipyramidal-shaped $Ag_5$ cluster. Therefore, it can be inferred that the interaction of methane plays a crucial role in stabilizing the silver AQC, even though it significantly affects the cluster's geometry. The calculated distance between the carbon atom and the nearest silver atom was determined to be 3.60 Å (see Table 2 for further details), which agrees well with a previous DFT prediction [32].

Additionally, the calculated Ag–Ag bond lengths of the cluster were nearly identical, with an average value of 2.64 Å, indicating the stability of the cluster. Furthermore, the physical adsorption of $CH_4$ was observed on the $Ag_5$ AQC with an energy value of $-0.50$ eV. Similarly, a small adsorption energy of methane has been found on $Ag_{20}$ [41] and platinum (Pt) [42] nanoclusters.

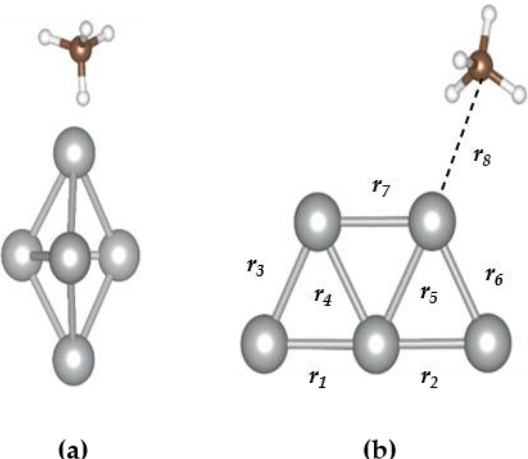

**Figure 3.** (**a**) Initial and (**b**) optimised structures of $CH_4@Ag_5$. Grey, brown, and white circles represent Ag, C, and H atoms, respectively.

**Table 2.** Bond lengths of $CH_4@Ag_5$ structure obtained using the PBE-DFT level.

| Symbol | Bond Length (Å) |
|---|---|
| $r_1$ | 2.62 |
| $r_2$ | 2.62 |
| $r_3$ | 2.63 |
| $r_4$ | 2.65 |
| $r_5$ | 2.65 |
| $r_6$ | 2.63 |
| $r_7$ | 2.65 |
| $r_8$ (Ag-C) | 3.60 |

In relation to charge transfer, the C atom's total electronic charge was determined to be 0.38 e$^-$. Of this charge, approximately 0.36 e$^-$ was transferred from the H atoms, while the remaining charge (~0.02 e$^-$) was transferred from the $Ag_5$ cluster, resulting in the oxidation of the cluster. The interaction between the $Ag_5$ cluster and the methane molecule was dominated by electrostatic forces, as evidenced by the charge density difference illustrated in Figure 4. The yellow clouds surrounding the Ag atoms indicate positive surface potentials, while the blue clouds on the $CH_4$ molecule indicate negative surface potentials. Based on the density of states' results, two electronic states were formed due to the contribution of the orbital of the C atom and the orbital of the H atoms. At approximately $-5.5$ eV, there was a clear overlap of the $CH_4$ orbitals (see Figure 4). Comparing the density of the states of the isolated $CH_4$ molecule (see Figure S3) with its density of states when interacting with the silver cluster, it can be observed that the electronic energy levels shifted towards lower energy levels by a factor of 3 eV. Notably, a weak hybridization of the methane molecule and the $Ag_5$ AQC occurred in the energy range of $-5$ eV to $-6$ eV, indicating a physical interaction between the methane and the silver cluster. This trend was also observed when adsorbing the $CH_4$ molecule on small tin-oxide clusters [43].

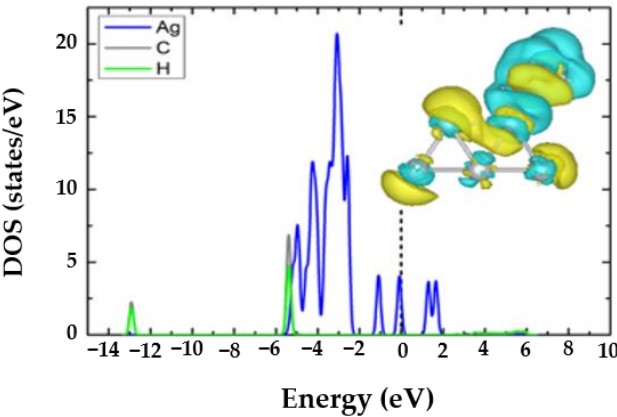

**Figure 4.** Charge density difference and density of states of $CH_4@Ag_5$. The blue, grey, and green curves of DOS represent the contributions of silver, carbon, and hydrogen orbitals, respectively. The vertical dotted line refers to the Fermi energy level.

### 3.3. The $H_2O@Ag_5$ Structure

Three different adsorption sites of $CO_2$ on the $Ag_5$ cluster have been explored to find the most stable structure (see Figures 5 and S8). It was found that the structure showed lower stability when the water molecule was adsorbed on the axial silver atom (see Figure S8d), while the structure revealed a higher stability when the water molecule was bonded to the equatorial silver atom (see Figures S8b and 5b). We further analysed the system where Figure 5 depicts the initial and optimized configurations of the most stable configuration of the $H_2O@Ag_5$ system. Initially, the $H_2O$ molecule was connected to three silver atoms (as shown in Figure 5a). Following geometrical optimization, as can be clearly seen in Figure 5b, the $H_2O$ molecule was attached to the $Ag_5$ AQC through an Ag–O bond of approximately 2.36 Å (see Table 3 for further details). This value for the Ag–O bond length is consistent with previous theoretical studies [44–47]. Notably, there was an insignificant change in the Ag–Ag bond length of the cluster after the adsorption of the water molecule. The calculated value for the $H_2O$ adsorption energy was approximately −0.50 eV, which suggests that the adsorption was physical rather than chemical. This adsorption energy value is in line with a previous study [48]. Table 4 illustrates the adsorption energies of the studied molecules on the $Ag_5$ AQC and compares the results with the literature.

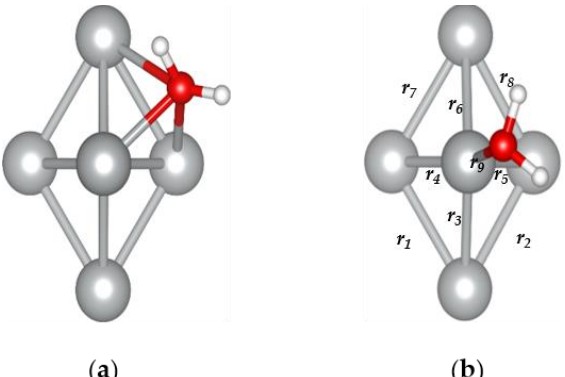

(**a**)  (**b**)

**Figure 5.** (**a**) Initial and (**b**) optimised structures of $H_2O@Ag_5$. Grey, red, and white circles represent Ag, O, and H atoms, respectively.

**Table 3.** Bond lengths of $H_2O@Ag_5$ structure obtained using the PBE-DFT level.

| Symbol | Bond Length (Å) |
|---|---|
| $r_1$ | 2.80 |
| $r_2$ | 2.80 |
| $r_3$ | 2.68 |
| $r_4$ | 2.72 |
| $r_5$ | 2.81 |
| $r_6$ | 2.80 |
| $r_7$ | 2.80 |
| $r_8$ | 2.80 |
| $r_9$ (Ag-*O*) | 2.36 |

**Table 4.** Adsorption energies ($E_{ads}$, eV) of the studied molecules on $Ag_5$ AQC compared with the literature.

| Structure | Present Study | Literature |
|---|---|---|
| $CO_2@Ag_5$ | −0.27 | −0.29 [32] |
| $CH_4@Ag_5$ | −0.50 | −0.27 [41] |
| $H_2O@Ag_5$ | −0.50 | −0.52 [48] |

Based on the Bader charge analysis, it can be inferred that the hydrogen atoms in the $H_2O$ molecule lose approximately 1.99 e$^-$, with the oxygen atom gaining around 1.92 e$^-$ and the remaining charge (~0.07 e$^-$) being transferred to the $Ag_5$ AQC, resulting in the reduction of the $Ag_5$ AQC. Further details on the charge distribution can be found in Table S7. The charge density difference and the density of the states of the $H_2O@Ag_5$ model are presented in Figure 6. Similar to the $CO_2@Ag_5$ and $CH_4@Ag_5$ systems, the $H_2O$ molecule exhibited negative surface potentials (represented by the blue clouds), while the silver atoms possessed positive surface potentials (represented by the yellow clouds), indicating that the interaction between the silver AQC and the $H_2O$ molecule is primarily driven by electrostatic forces. The density of the states information in Figure 6 reveals that the water molecule produced four peaks due to the bonds between the hydrogen atoms and the oxygen atom. Specifically, the electronic state appearing at approximately −9.8 eV was mainly derived from the notable mixture of hydrogen and oxygen orbitals, resulting in the charge transfer from the hydrogen atoms to the oxygen atom. Comparing the density of states of the isolated water molecule (see Figure S5) with its density of states when interacting with the $Ag_5$ cluster (see Figure 6), a clear shift in the energy level states of the water molecule towards lower energy levels can be observed, primarily due to the interaction with the $Ag_5$ AQC. Additionally, the Ag–O bond led to the formation of an overlapped electronic state whose peak was located at approximately −6.3 eV, providing evidence for the charge transfer between the $Ag_5$ cluster and the water molecule.

In pursuit of the objectives outlined in this work, it is important to note that the structures presented herein may not represent the most stable configurations. Therefore, an additional step is required to identify the most stable structures, which would significantly enhance the theoretical predictions of the electronic structures of the molecules of $Ag_5$ systems. Moreover, the selection of exchange-correlation functional presents an obvious challenge, and the application of hybrid functional, such as HSE06 [48], could lead to more accurate electronic property predictions. To precisely determine and validate the amount of charge transfer between species, alternative methods of charge transfer analysis, such as Hirshfeld [49], Voronoi [50], or Mulliken [51], could be employed.

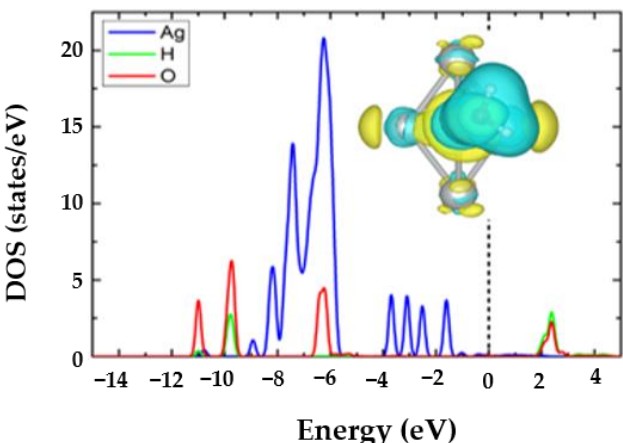

**Figure 6.** Charge density difference and density of states of $H_2O@Ag_5$. The blue, red, and green curves of DOS represent the contributions of silver, oxygen, and hydrogen orbitals, respectively. The vertical dotted line refers to the Fermi energy level.

### 4. Concluding Remarks

In the present study, we have investigated the interactions between the $Ag_5$ AQC and various molecules, namely methane, carbon dioxide, and water, utilizing DFT predictions. Specifically, we have employed Bader charge analysis to examine the charge transfer between the $Ag_5$ cluster and the molecules. Our results reveal that the $Ag_5$ cluster experiences slight charge loss when interacting with $CO_2$ and $CH_4$ molecules. Conversely, the cluster gains electrons upon interaction with the $H_2O$ molecule. Furthermore, we have performed adsorption energy calculations to assess the strength of the adsorption of water, methane, and carbon dioxide on the $Ag_5$ AQC. Our results indicate that both water and methane exhibit stronger adsorption on the $Ag_5$ AQC compared to the carbon dioxide molecule, with an energy difference of 0.23 eV. The atomic arrangement of the $CO_2@Ag_5$ compound involves a singular oxygen atom covalently bonded to a singular silver atom, with a bond length measuring 2.56 Å. The analysis reveals that the distance between the carbon atom and the closest silver atom is 3.25 Å, which aligns with the findings of a prior DFT investigation. Conversely, structures lacking a bond between the oxygen atom and the silver atom exhibit lower stability. Notably, we have observed a significant deformation of the $Ag_5$ AQC following its interaction with the methane molecule. This deformation suggests that the $Ag_5$ AQC could potentially be further stabilised by introducing $CH_4$ into the system. In the case of the $H_2O@Ag_5$ system, it was noted that the $H_2O$ molecule was connected to the $Ag_5$ AQC via an Ag–O bond spanning approximately 2.36 Å. Remarkably, the Ag–Ag bond length of the cluster experienced minimal alteration following the adsorption of the water molecule. The computed $H_2O$ adsorption energy was approximately −0.50 eV, indicating a predominantly physical, rather than chemical, adsorption process. The results in this work can be significantly enhanced by implementing hybrid functionals to gain a deeper understanding of the electronic behaviour of the studied molecules when interacting with the $Ag_5$ AQC. To sum up, our theoretical predictions offer valuable insights into the interactions between the $Ag_5$ AQC and $CO_2$, $CH_4$, and $H_2O$ molecules. These results hold potential significance for future experimental studies, providing a foundation for further exploration in the field of renewable energy.

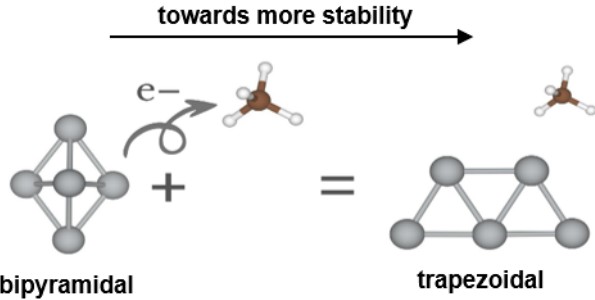

For Table of Contents Only

**Supplementary Materials:** The following supporting information can be downloaded at: https://www.mdpi.com/article/10.3390/cryst13121691/s1, Figure S1. Optimised structures of isolated (a) $CO_2$, (b) $CH_4$, (c) $H_2O$, and (d) $Ag_5$ systems. Note: numbers indicate bond length values. Table S1. Charge distribution of $CO_2$ molecule. Table S2. Charge distribution of $CH_4$ structure. Table S3. Charge distribution of $H_2O$ molecule. Table S4. Charge distribution of $Ag_5$ structure. Figure S2. Density of states of isolated $CO_2$ molecule. Figure S3. Density of states of isolated $CH_4$ molecule. Figure S4. Density of states of bare $Ag_5$ cluster. SOMO: singly occupied molecular orbital. Figure S5. Density of states of isolated $H_2O$ molecule. Table S5. Charge distribution of $CO_2@Ag_5$ structure. Table S6. Charge distribution of $CH_4@Ag_5$ structure. Table S7. Charge distribution of $H_2O@Ag_5$ structure. Figure S6 Different adsorption sites of $CO_2@Ag_5$. Table S8. Total energy of the structures presented in Figures S6 and 1. Figure S7 Different adsorption sites of $CH_4@Ag_5$. Table S9. Total energy of the structures presented in Figures S7 and 3. Figure S8 Different adsorption sites of $H_2O@Ag_5$. Table S10. Total energy of the structures presented in Figures S8 and 5. Figure S9 Initial (a), (c), and (e) and optimised (b), (d), and (f) structures of $CO_2@Ag_5$, $CH_4@Ag_5$, and $H_2O@Ag_5$, respectively using a cut-off energy value of 50 Ry.

**Author Contributions:** M.A. (Moteb Alotaibi) originally conceived the concept; the calculations were carried out by M.A. (Moteb Alotaibi), M.A. (Majed Alshammari), and T.A. All authors provided essential contributions to interpreting the data reported in this manuscript. A.K.I. coordinated the writing of the manuscript with input from M.A. (Moteb Alotaibi), M.A. (Majed Alshammari), and T.A. All authors have read and agreed to the published version of the manuscript.

**Funding:** This project was funded by the Deanship of Scientific Research at Prince Sattam bin Abdulaziz University award number 2023/01/26562, (Alkharj, Saudi Arabia).

**Data Availability Statement:** Data are contained within the article.

**Acknowledgments:** Moteb Alotaibi is grateful to the Deanship of Scientific Research at Prince Sattam bin Abdulaziz University, Alkharj. Moteb Alotaibi and Majed Alshammari are thankful for computer time, this research used the resources of the Supercomputing Laboratory at King Abdullah University of Science & Technology (KAUST) in Thuwal, Saudi Arabia. Majed Alshammari and Turki Alotaibi are grateful to Jouf University (Saudi Arabia). Ali Ismael is grateful for financial assistance from Lancaster University, UK.

**Conflicts of Interest:** The authors declare no conflict of interest.

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
