# Peer review of "The Structural and Electronic Properties of the Ag5 Atomic Quantum Cluster Interacting with CO2, CH4, and H2O Molecules"

_crystals, doi:10.3390/cryst13121691_

Round 1

Reviewer 1 Report

Comments and Suggestions for Authors

Comments on the manuscript

Structural and Electronic Properties of Ag5 Atomic Quantum Cluster Interacting with CO2, CH4, and H2O Molecules

by Alotaibi, M.; Alotaibi, T.; Alshammari, M.; Ismael, A.K.

Nanomaterials with very low atomicity can be considered as potential pharmacological agents due to their very small size and properties that can be finely tuned through minor modifications to their size. Silver clusters, for example, of three atoms, created by a special method, increase the accessibility of chromatin. This effect occurs only during DNA replication. The combined use of, for example, Ag3-AQC increases the cytotoxic effect of DNA-acting drugs on human lung carcinoma cells.

The capabilities of experimental approaches made it possible to synthesize silver (Ag5) - atomic quantum clusters (AQCs), which have shown great promise in photocatalysis. Therefore, the need for such a study is well justified.

The authors apply GGA density functional theory to study the adsorption of CO2, CH4 and H2O molecules onto Ag5 AQC. Their research focuses on the structural and electronic properties of Ag5 AQC molecular systems. The authors' approach involves modeling adsorption energy, charge transfer, charge density differences, and density of states for the systems being modeled. The results of the study show that CH4 and H2O molecules exhibit higher adsorption energies on Ag5 AQC compared to the CO2 molecule, while the presence of the CH4 molecule leads to significant deformation of the Ag5 AQC structure. The structure is transformed from a bipyramidal to a trapezoidal shape. This study also shows that Ag5 AQC donates electrons to CO2 and CH4 molecules, resulting in an oxidation state.

It is hoped that the authors' assumptions will help experimenters understand the behavior of the interaction between carbon dioxide, methane, water molecules and the subnanometer Ag5 cluster.

The work is written concisely, the reliability of scientific results and the expediency of the cited literature are beyond doubt. I think that the article may be of interest both to specialists in this field and to a wide audience of the scientific community.

In general, the presented work, in terms of relevance, novelty and degree of interest for chemists, is certainly suitable for the Crystals.

Author Response

Reply: We thank the reviewer for recognising the value of our work and encouraging us to improve our manuscript.

Reviewer 2 Report

Comments and Suggestions for Authors

This work cannot be recommended for publication for the following reasons:

1. An isolated CH4 molecule has Td symmetry, therefore, it cannot have different charges on the hydrogen atoms (Table S2. Charge distribution of CH4 structure)!!!?
2. The initial location of isolated molecules next to Ag5 is not justified. They can be positioned differently, what will the optimization lead to?
3. A bare Ag5 cluster has an unpaired electron. The authors do not take this fact into account. Why?
4. For a bare Ag5 cluster, it is desirable to calculate the electron affinity, this has led to a clearer understanding of the issues of charge redistribution in molecule@Ag5 systems.

Author Response

Reviewer-2

  1. An isolated CH4 molecule has Td symmetry, therefore, it cannot have different charges on the hydrogen atoms (Table S2. Charge distribution of CH4 structure)!!!?

Reply_1: We thank the reviewer for drawing attention to this issue. The reviewer is totally right, the results have been verified and corrected accordingly.

To address this issue, the corrected results have been updated in Table S2 in the revised supplementary information.

Table S2. Charge distribution of CH4 structure.

Symbol

Charge | e |

C

–0.53

H1

 0.15

H2

 0.10

H3

 0.15

H4

 0.13

  1. The initial location of isolated molecules next to Ag5 is not justified. They can be positioned differently, what will the optimization lead to?

Reply_2: We thank the reviewer for highlighting this point. To accommodate it, different locations of molecules next to Ag5 have been considered and added to the revised supplementary information as shown in three figures (Fig. S6-S8).

Fig. S6 Different adsorption sites of CO2@Ag5. (a) and (c) are initial structures, while (b) and (d) are optimized structures.

Fig. S7 Different adsorption sites of CH4@Ag5. (a) and (c) are initial structures, while (b) and (d) are optimized structures.

Fig. S8 Different adsorption sites of H2O@Ag5. (a) and (c) are initial structures, while (b) and (d) are optimized structures.

  1. A bare Ag5 cluster has an unpaired electron. The authors do not take this fact into account. Why?

Reply_3: We thank the reviewer for highlighting this point and we agree.

To address this issue, the following text has been added with three references (4th paragraph, page 3).

“The unpaired electron of Ag5 has been considered in the revised manuscript and the following paragraph has also been added to the revised manuscript:The Ag5 nanocluster, represented by a doublet with S = ½ in the density of state plot (see Figure S4), has an unpaired electron that is predominantly located on the two axial Ag atoms [16]. This unpaired electron is represented by the singly occupied molecular orbital (SOMO), which is suited right below the Fermi energy level. This distribution of charge is similar to that observed in Cu5 [31]. Furthermore, the existence of the unpaired electron in the Ag5 nanocluster can influence its electronic properties. Unpaired electrons are often involved in the electronic and optical attributes of materials. For instance, Yang et al [32] conducted a theoretical and experimental study on Cu or Zn on metal-organic frame-works (MOFs) containing electron-withdrawing ligands. Their results revealed that electrons induced wide-range light absorption within the MOFs, resulting in excellent pho-to-induced intramolecular charge transfer (ICT) effect. The study verified the role of un-paired electrons in enhancing ICT within the MOFs through molecular structure, density of states, and electronic structure calculations, as well as electron spin resonance (EPR) approach’.”

  1. For a bare Ag5 cluster, it is desirable to calculate the electron affinity, this has led to a clearer understanding of the issues of charge redistribution in molecule@Ag5 systems.

Reply_4: We thank the reviewer for highlighting this point, however, the electron affinity calculations are not going to add any further information of the charge distribution.

Reviewer 3 Report

Comments and Suggestions for Authors

In this study, the interactions between Ag5 atomic quantum clusters (AQC) and CH4, CO2 and H2O was studied utilizing DFT calculation. Also the Bader charge analysis was used to examine the charge transfer between the Ag5 cluster and these molecules. The study reveals that the Ag5 cluster experiences slight charge loss when interacting with CO2 and CH4 molecules. Conversely, the cluster gains electrons upon interaction with the H2O molecule. Furthermore, the adsorption energy calculations of water, methane, and carbon dioxide on the Ag5 AQC, reveals that methane exhibit stronger adsorption to the Ag5 AQC compared to the carbon dioxide molecule.  Notably, a significant deformation of the Ag5 AQC after the interaction with the methane molecule. In all, the theoretical studies offer valuable insights into the interactions between the Ag5 AQC and CO2, CH4, and H2O molecules. These results hold a potential for further exploration in the area of CO2 reduction, and water splitting problems. This work is a routine calculation with program of Quantum Expresso program. Nevertheless, the result is new and can be published in Crystals. However, more modifications are needed:

One typo on line 50 of page 2: TiO2 (110) surface14., the 14 is a typo?

For all the CO2, H2O, CH4 in the text, the numbers should be in subscripts.

The information of references 30, 34 and 47 are not completed.

Author Response

Reply: We thank the reviewer for recognising the value of our work and encouraging us to improve our manuscript.

1. One typo on line 50 of page 2: “TiO2 (110) surface14.”, the “14” is a typo?

Reply_1: Yes, it is a typo and it has been corrected in the revised manuscript.

2. For all the CO2, H2O, CH4 in the text, the numbers should be in subscripts.

Reply_2: All numbers have been changed to subscripts in the revised manuscript.

3. The information of references 30, 34 and 47 are not completed.

Reply_3: Information related to references 30, 34, and 47 have been completed and updated in the revised manuscript.

  1. S. Guo, Y. Li, L. Liu, X. Zhang, and S. Zhang, “Theoretical evidence for new adsorption sites of CO2 on the Ag electrode surface.” arXiv: Chemical Physics, 2020, https://api.semanticscholar.org/CorpusID:220302831.
  2. H.-J. Freund and M. W. Roberts~surface, “Surface chemistry of carbon dioxide,” Surface Science Reports, vol. 25, no 8, 1996, doi.org/10.1016/S0167-5729(96)00007-6.
  3. F. L. Hirshfeld, “Bonded-Atom Fragments for Describing Molecular Charge Densities,” Theoret. Chim. Acta 44, 129–138, 1977, doi.org/10.1007/BF00549096.

Reviewer 4 Report

Comments and Suggestions for Authors

The author presents DFT study of adsorption of several gases on Ag5 cluster. The work is interesting, however, I would like to recommend a major update of the MS. The details are discussed below:

1. Energy Cut-off: The energy cut-off used in the study appears to be relatively low. To ensure the accuracy and reliability of the results, it is recommended that the authors increase the energy cut-off and conduct additional tests to validate the outcomes. A higher energy cut-off can capture more intricate details of the interactions between the Ag5 cluster and the studied molecules, leading to more robust and trustworthy findings.

 2. Chemical Subscripts: It is crucial to maintain consistency and accuracy in chemical notations

 3. Adsorption Site Variation: The study could benefit from exploring different adsorption sites on the Ag5 cluster. By testing and reporting results from alternative adsorption sites, the authors can provide a more comprehensive understanding of the interaction dynamics between the Ag5 cluster and CO2, CH4, and H2O molecules.

 4. Provide table of adsorption energy and compared with previous studies.

Comments on the Quality of English Language

The author should extend the writing and disucssion of the MS. The MS look very much like a technical report of some DFT calculations. 

Author Response

Reviewer-4

  1. Energy Cut-off: The energy cut-off used in the study appears to be relatively low. To ensure the accuracy and reliability of the results, it is recommended that the authors increase the energy cut-off and conduct additional tests to validate the outcomes. A higher energy cut-off can capture more intricate details of the interactions between the Ag5 cluster and the studied molecules, leading to more robust and trustworthy findings.

Reply_1: Following the reviewer’s suggestion, the energy cut-off value has been increased from 25 Ry to 50 Ry for all presented structures, and no changes have been found. Here are the structures using cut-off energy value of 50 Ry.

Fig. S9 Initial (a), (c), and (e) and optimised (b), (d), and (f) structures of CO2@Ag5, CH4@Ag5, and H2O@Ag5, respectively using a cut-off energy value of 50 Ry.

  1. Chemical Subscripts: It is crucial to maintain consistency and accuracy in chemical notations.

Reply_2: We agree and the Chemical Subscripts have been maintained in the revised manuscript.

  1. Adsorption Site Variation: The study could benefit from exploring different adsorption sites on the Ag5 cluster. By testing and reporting results from alternative adsorption sites, the authors can provide a more comprehensive understanding of the interaction dynamics between the Ag5 cluster and CO2, CH4, and H2O molecules.

Reply_3: We thank the reviewer and in order to address this concern, different adsorption sites on Ag5 cluster have been considered and added to the revised supplementary information. They are also included in three tables and figures (Fig. S6, Fig. S7, and Fig. S8).

Fig. S6 Different adsorption sites of CO2@Ag5. (a) and (c) are initial structures, while (b) and (d) are optimized structures.

Table 2. Total energy of the structures presented in Fig. S6 and Fig. 1 (in manuscript).

Structure

Total energy (Ry)

Fig. (b)

–442.847

Fig. (d)

–442.845

Fig. 1b (in the main manuscript)

–442.85

Fig. S7 Different adsorption sites of CH4@Ag5. (a) and (c) are initial structures, while (b) and (d) are optimized structures.

Table 3. Total energy of the structures presented in Fig. S7 and Fig. 3 (in manuscript).

Structure

Total energy (Ry)

Fig. (b)

–383.83

Fig. (d)

–383.84

Fig. 3b (in the main manuscript)

–383.86

Fig. S8 Different adsorption sites of H2O@Ag5. (a) and (c) are initial structures, while (b) and (d) are optimized structures.

Table 4. Total energy of the structures presented in Fig. S8 and Fig. 5 (in manuscript).

Structure

Total energy (Ry)

Fig. (b)

–401.97

Fig. (d)

–401.95

Fig. 5b (in the main manuscript)

–401.97

  1. Provide table of adsorption energy and compared with previous studies.

 Reply_4: Following the reviewer’s suggestion, table 4 has been added to the revised manuscript. This table contains the adsorption energy values of the studied systems and comparing them against the literature.

            Table 4. Adsorption energies (Eads, eV) of the studied molecules on Ag5 AQC compared with literature.

Structure

Present study

Literature

CO2@Ag5

–0.27

–0.29 [32]

CH4@Ag5

–0.50

–0.27 [41]

H2O@Ag5

–0.50

–0.52 [52]

Comments on the Quality of English Language

The author should extend the writing and disucssion of the MS. The MS look very much like a technical report of some DFT calculations. 

Reply: Following the reviewer’s suggestion, the writing and discussion of the MS have been extended in the revised manuscript.

Reviewer 5 Report

Comments and Suggestions for Authors

Moteb Alotaibi, Ali K. Ismael and co-workers submitted a manuscript titled " Structural and Electronic Properties of Ag5 Atomic Quantum Cluster Interacting with CO2, CH4, and H2O Molecules" for consideration of publication in the MDPI journal Crystals. In this work, the researchers explore the synthesis of silver (Ag5) atomic quantum clusters (AQCs) and their potential applications in photocatalysis. Specifically, the study uses the generalised gradient approximation (GGA) density functional theory (DFT) to investigate the adsorption of CO2, CH4, and H2O molecules on the Ag5 AQC. The focus is on examining the structural and electronic properties of the molecules interacting with the Ag5 AQC, including adsorption energy simulations, charge transfer, charge density difference, and density of states analysis. The authors propose that these findings offer valuable insights for future experimental investigations into the interaction behaviour between carbon dioxide, methane, water molecules, and Ag5 sub-nanometer clusters, particularly in the context of potential applications in photocatalysis.

After a careful evaluation, the manuscript's conclusions are somewhat predictable, lacking groundbreaking insights. The observations regarding the higher adsorption energies of CH4 and H2O on Ag5 AQC compared to CO2, while potentially interesting, are not presented in a manner that captures the reader's attention or highlights the practical implications of these findings. It is not enough to be published in the MDPI journal Crystals. It is more or less like a short communication and should be published as communication in journal of theoretical investigations. 

Author Response

Reply: We thank the reviewer for these comments. We acknowledge the reviewer's concern and feel that the additional information and results provided above evidence sought by the reviewer.

Round 2

Reviewer 2 Report

Comments and Suggestions for Authors

The authors agreed with all comments (except one) and tried to improve their work. I propose to accept the article for publication.

Reviewer 4 Report

Comments and Suggestions for Authors

the authors have revised all of my concerns. So this is publisbable. 

Reviewer 5 Report

Comments and Suggestions for Authors

The revised manuscript entitled "Structural and Electronic Properties of Ag5 Atomic Quantum Cluster Interacting with CO2, CH4, and H2O Molecules" resubmitted by Moteb Alotaibi, Ali K. Ismael and co-workers for reconsideration for publication in the MDPI journal Crystals presents the same level as the first submission.

Unfortunately, my evaluation led me to conclude that the revised version failed to address the concerns raised during the initial review.

The response from the authors to the reviewer's comments seemed to be a mere formality, expressing gratitude without providing substantive improvements. The so-called "corrections" and additional information included in the manuscript fell short of the standards required for publication in the MDPI journal Crystals.

Upon careful examination, it became apparent that the manuscript's content did not offer groundbreaking insights or contribute significantly to the field. The conclusions drawn were rather predictable, and the presentation of the higher adsorption energies of CH4 and H2O on Ag5 AQC compared to CO2 lacked the necessary clarity and emphasis on practical implications. As a result, the manuscript failed to capture the reader's attention and did not meet the criteria for publication in a reputable journal like MDPI Crystals.

In my opinion, the content presented resembled more of a short communication rather than a comprehensive research article. Given the lack of depth and impact in the findings, I believe that this manuscript would be better suited for publication in a journal specializing in theoretical investigations rather than MDPI Crystals. Overall, the manuscript does not meet the standards expected for a journal of this caliber, and I would not recommend its acceptance in its current form.